# Predictive Factors for Failure of Intraarticular Injection in Management of Adhesive Capsulitis of the Shoulder

**DOI:** 10.3390/jcm11206212

**Published:** 2022-10-21

**Authors:** Stefan J. Hanish, Mathew L. Resnick, Hyunmin M. Kim, Matthew J. Smith

**Affiliations:** Department of Orthopaedic Surgery, University of Missouri School of Medicine, Columbia, MO 65212, USA

**Keywords:** adhesive capsulitis, frozen shoulder, corticosteroid, injection, failure, diabetes mellitus, hypothyroid

## Abstract

Intraarticular (IA) corticosteroid injections have been demonstrated to be an effective management for adhesive capsulitis in both the short- and mid-term. Yet, certain patients fail to improve both subjectively and clinically. This study aims to identify predictive factors for treatment failure of IA injections in management of adhesive capsulitis. A retrospective review found 533 patients undergoing IA corticosteroid or IA NSAID injection for adhesive capsulitis between June 2015 and May 2020 at a single healthcare institution. Patient demographics characteristics, comorbidities, pain scores, and range of motion were compared. Treatment failure was defined as need for subsequent IA injection within 6 months or progression to surgical management within 12 months. 152 patients (28.52%) experienced treatment failure of IA corticosteroid injection. Pre-injection pain scores were greater for those who experienced treatment failure (5.40 vs. 4.21, *p* < 0.05). Post-injection pain scores were greater for those who experienced treatment failure (3.77 vs. 2.17, *p* < 0.01). Reduced post-injection external rotation in abduction also predicted treatment failure (56.88° vs. 70.22°, *p* < 0.01). IA corticosteroid injections are associated with increased rates of failure and progression to surgical management when patients present with increased pain levels as well as with less improvement in pain levels and ROM following injection.

## 1. Introduction

Adhesive capsulitis of the glenohumeral joint, also known as “frozen shoulder,” is a common pathology characterized by pain and progressive range of motion restriction [1,2,3,4]. The diagnosis is made clinically and defined as shoulder pain with restricted movement in both active and passive range of motion, with normal radiographic scans of the glenohumeral joint [5]. Despite being widely regarded in literature as a self-limiting pathology, many investigators have reported that complete resolution is not guaranteed [5,6,7]. Approximately 40–50% of patients experience persistent symptoms [5,6,8,9], and 10–15% of patients require surgical management [6,9,10,11].

Consensus exists amongst clinicians and investigators that initial management for adhesive capsulitis should be non-surgical. Current literature describes a myriad of available treatment modalities including physical rehabilitation, oral NSAIDs, oral glucocorticoids, intraarticular (IA) corticosteroid injection, hyaluronic acid injections, and/or hydrodilation (arthrographic distension). Further, high-level evidence directly comparing treatment modalities is scarce [12,13,14,15].

Intra-articular corticosteroid injections, have been shown by numerous studies to provide significant benefit in function and pain in the short-term [1,5,16,17,18,19,20,21]. However, current literature demonstrates that 5–15% of patients do not experience the functional and pain-reducing benefits of IA corticosteroid injections [16]. Certain comorbidities have been associated with reduced benefit from steroid injections such as diabetes, which has been associated with soft tissue thickening and proliferation as well as greater deficits in range of motion [16]. The resulting scarcity of strong evidence coupled with the unclear pathophysiology of the disease can create a treatment dilemma for healthcare professionals [1].

The aim of this study is to identify comorbidities and risk factors associated treatment failure of IA injections for the management of adhesive capsulitis. We hypothesize that increased pre-injection pain level, greater pre-injection ROM restriction, and history of diabetes will result in increased rates of treatment failure of IA corticosteroid injection.

## 2. Materials and Methods

After approval from institutional review board was obtained, a retrospective case–control review was performed for all patients treated for adhesive capsulitis at our institution between June 2015 and May 2020. All information was requested utilizing REDCap to search the electronic medical record (EMR), including the EMR data from other participating institutions. Inclusion criteria were patients with a diagnosis of adhesive capsulitis of the shoulder with adequate follow-up data including pain scores and range of motion. Exclusion criteria included absence of follow-up data recorded. Potential candidates for inclusion were identified using ICD-9 code (726.0), ICD-10 codes (M75.00, M75.01, M75.02), CPT codes (20610, 20611), and HCPCS codes (J3301, J1030, J1885).

A total of 533 patients were included in the study. A diagnosis of adhesive capsulitis was defined as a clinical diagnosis made by a physician based on patient history involving pain and loss of both active and passive range of motion without acute injury and normal radiographs. Each included patient received an intra-articular injection. Intra-articular injections were administered with ultrasound-guidance or landmark-guidance based on physician experience and preference. For a small group of 10 patients for whom corticosteroids were contraindicated based on allergy or brittle diabetes, IA nonsteroidal anti-inflammatory drugs (NSAIDs) were chosen. This subset was too small for separate sub group analysis and were included in the injection group as a whole. Intra-articular injections were standard dosages used at our institution including 10 mg triamcinolone acetonide, 40 mg methylprednisolone acetate, or 15 mg ketorolac tromethamine.

Each chart was individually reviewed for data collection (SH, MR). Demographic data recorded included age, sex, race, and BMI. The presence of concurrent medical comorbidities was also documented by review of primary care physician notes, which included diabetes mellitus, hypothyroidism, hyperthyroidism, and smoking history. Descriptors of the disease course of adhesive capsulitis were recorded from orthopedic clinic, primary care, and physical therapy notes including dates of diagnosis, injection, and follow-up visits, laterality of disease, pre-injection VAS pain score and range of motion, and post-injection VAS pain score and range of motion during the initial 3 months after the injection. Range of motion was measured by clinical assessment or by goniometer in external rotation in abduction, external rotation at the side, and forward elevation. External rotation in abduction and external rotation at the side were classified as mild (>45°), moderate (20–45°), or severe restriction (<20°). Forward elevation was classified as mild (>120°), moderate (90–120°), or severe (<90°). Procedure details regarding choice of drug, dosage, and use of ultrasound-guidance or landmark-guidance were collected. Patients underwent concurrent formal standardized physical therapy at our institution or were provided recommendations for home physical therapy exercises if formal physical therapy was declined.

Treatment failure was defined as the need for an unplanned, subsequent IA injection within 6 months of previous IA injection or progression to surgical management (e.g., arthroscopic capsular release, lysis of adhesions, manipulation under anesthesia) within 12 months of previous IA injection. Indications for subsequent injection and progression to surgical management were multivariable in nature after incomplete resolution of symptoms following injection. Considerations in this decision included the physician’s assessment of the patient’s response to the initial IA injection, severity of symptoms, patient satisfaction with function, and patient desire to progress to more invasive levels of care. Indications for progression to surgery included physician assessment and symptoms refractory to IA corticosteroid.

Extracted data was analyzed by statisticians within our institution. Categorical variables are reported in frequencies and percentages. For demographics, comorbidities, and range of motion categorical variables, chi-square tests for association or fisher exact test were used. In addition, statistical analysis was conducted using independent two-tailed t-test or Mann–Whitney to determine significant associations between patient-based variables and treatment failure of IA injections. A *p*-value of 0.05 was deemed statistically significant.

## 3. Results

Patient demographic and clinical examination data is summarized in Table 1 and Table 2. The majority of patients undergoing IA corticosteroid for management of adhesive capsulitis were female (274 patients, 66.34%), mean age was 55.34 ± 10.67 years (range, 20–86), and mean BMI was 30.94 ± 7.84 kg/m^2^ (range, 14–64) (Table 1). The majority of the patients were Caucasian (368 patients, 89.32%). Comorbidities within the populations included diabetes mellitus (129 patients, 31.31%) and thyroid disease (73 patients, 13.70%).

A total of 152 patients (28.52%) experienced treatment failure of IA corticosteroid injection, as defined by subsequent IA corticosteroid injection within 6 months or progression to surgery. A total of 115 (21.58%) received a repeat injection within 6 months and 48 (9.01%) progressed to surgery. A total of 11 (2.06%) both received a repeat injection within 6 months and progressed to surgery.

Between those who experienced treatment failure, there were no differences in sex, mean age, mean BMI), or proportion of patients with diabetes or thyroid (*p* > 0.05).

The ROM of each study participant in external rotation in abduction, external rotation at the side, and forward elevation was categorized by severity as mild, moderate, or severe (Table 3). The relationship between severity of ROM restriction and treatment failure in this patient population is displayed in Table 4. Post-injection external rotation in abduction was significantly reduced for those who experienced treatment failure versus those who did not (56.88° vs. 70.22°, *p* = 0.0068). Other than this, there were no other significant associations between treatment failure and pre-injection or post-injection ROM in external rotation in abduction, external rotation at the side, or forward elevation (*p* > 0.05).

Pre-injection and post-injection (at approximately 3 months) pain levels were measured using VAS pain score. Patients who experienced treatment failure of IA injection had significantly higher pre-injection pain scores (5.40 vs. 4.21, *p* = 0.0203) and post-injection pain scores (3.77 vs. 2.68, *p* = 0.002) versus those who did not experienced treatment failure.

## 4. Discussion

IA corticosteroid injections have been proven to benefit patients with adhesive capsulitis in the short and mid-term by reducing pain and improving function [1,5,17,18,19,20,21]. Yet, some patients do not show subjective or clinical improvement with one or more IA corticosteroid injections. Few studies exist that have investigated predictive factors for treatment failure in these treatment-resistant cohorts. This current study supports our original hypothesis by demonstrating that IA corticosteroid injections for the management of adhesive capsulitis are associated with increased rates of failure and progression to surgical management when patients present with increased pain levels and experience less improvement in pain levels and ROM (e.g., ER in abduction) following injection. The association of higher pre-injection pain scores with treatment failure is a clinically impactful finding as it may allow physicians to provide improved patient-specific counseling as to likelihood of treatment success. It may improve patient expectations and decisions making when the injections do not hasten resolution.

Adhesive capsulitis primarily involves contracture of the joint capsule and the rotator interval, composed of the superior glenohumeral ligament and the coracohumeral ligament [18,22]. Frozen shoulder classically transitions through three separate stages including “freezing” (gradually worsening pain and range of motion for 2–9 months), “frozen” (severe range of motion restriction for 4–12 months), and “thawing” (gradual improvement in shoulder motion over 5–24 months) [1]. However, at this time, the specific pathophysiology remains unclear. Current models suggest that chronic inflammation within the capsular subsynovial layer eventually causes progression to capsular fibrosis, contracture, and adherence to the anatomic neck of the humerus [4,18,23,24,25]. Several inflammatory cytokines have been implicated in the development of this inflammatory contracture, such as transforming growth factor beta (TGF-b), tumor necrosis factor alpha (TNF-a), and interleukins [18]. Determining the pathogenesis of adhesive capsulitis is of great importance as the pathophysiology will guide selection of future treatment modalities that can improve objective and subjective measures in the disease course [5]. As adhesive capsulitis often resolves without surgical intervention, a prolonged course of nonoperative management is offered in an attempt to avoid surgical intervention. While earlier studies painted a controversial picture regarding the benefit of IA corticosteroids in the management of adhesive capsulitis, recent studies have provided convincing evidence that supports their use [1,5,16,17,18,19,20,21] Investigators have demonstrated IA corticosteroid injections to produce effective pain relief and functional improvements in the short- and mid-term [1,5,17,18,19,20,21,26]. A 2020 meta-analysis of treatment modalities for adhesive capsulitis recommended the use of IA corticosteroid for patients with frozen shoulder of duration less than 1 year as it provides earlier benefits than other interventions and is associated with superior outcomes compared with no treatment for all outcome measures analyzed [1]. The recommendation includes an accompanying physical therapy program which, in tandem with IA corticosteroid, produced superior benefits than IA corticosteroid alone [1]. This current study supports these findings showing improvements in VAS pain scores and range of motion at 3 months post-injection. However, despite proven benefit of IA injections, current literature emphasizes the short-term nature of the benefit [5,17,18,19,20,21]. Improvements in pain and function have typically been shown to last for a maximum of 24 weeks with diminishing efficacy compared to control groups thereafter [5,17,18,19,20,21,26,27,28,29,30,31,32,33,34]. This demonstration of significant benefit from IA steroids in recent literature has impactful ramifications for general and specialist musculoskeletal practitioners, as it provides them with an accessible, cost-effective, and evidence- based treatment to supplement the classical conservative management of physical therapy [1].

There is a paucity of existing literature that describes predictive factors of treatment failure of IA corticosteroids for the management of adhesive capsulitis. In the meager of amount of investigation that has been performed, studies focus on attempting to find demographic factors and comorbidities that predict treatment failure [9,24,34,35,36]. A comprehensive 2021 meta-analysis by Zhang et al. displayed a deleterious impact on conservative management with IA corticosteroids by female sex and diabetes [36]. This current study found no correlation between female sex or diabetes with increased rates of treatment failure. The meta-analysis by Zhang et al. likely benefitted from superior power due to larger sample size to identify these correlations. While few studies have sought comorbidities and patient demographics as predictor factors for failure, even fewer have investigated associations between treatment failure and clinical measurements, such as restricted range of motion or pain [9]. Ruiz et al. explored the utility of precise assessment of isolated glenohumeral ROM in patients with primary adhesive capsulitis and its ability to help identify patients in whom conservative measures might fail [9]. Results showed that those who did not experience early functional improvement in isolated glenohumeral ROM often required surgery [9]. No other study has explored or demonstrated a relationship between pain level and failure of IA corticosteroid injections. This current study found an association between increased rates of treatment failure of IA corticosteroid with elevated pre-injection and post-injection pain scores as well as greater restrictions of external rotation in abduction. This is significant as it is the first time that an association of pre-injection and post-injection pain has been directly associated with increased likelihood of treatment failure. To our knowledge, it is also the first time that restricted range of a specific glenohumeral joint motion (ER in abduction) has been associated with greater rates of treatment failure. Determining predictive factors for the failure of IA corticosteroids will allow clinicians to make evidence-based decisions regarding management of adhesive capsulitis in specific populations.

This current study found no significant differences in patient demographic features or medical comorbidities for treatment failure of IA corticosteroid in adhesive capsulitis. The prevalence of adhesive capsulitis in patients with a diagnosis of diabetes mellitus has been documented to be approximately 20%, compared to 2–5% in the general population [17]. This association with diabetes has been attributed to its association with soft tissue thickening and proliferation, which has been speculated to cause increased range of motion restriction in diabetics with adhesive capsulitis [16]. Erickson et al. showed that patients with diabetes respond less favorably to intra-articular injections [16]. Yet, other studies including Roh et al., have shown that intra-articular injections in diabetics show superior benefit as compared to home physical therapy [37]. Due to the paucity of high-level literature describing demographic features and medical comorbidities associated with treatment failure of intra-articular injections, more research is warranted on this topic.

This study has several limitations. First, it is a retrospective cohort analysis, and, though the number of included patients is high, it suffers from the limitations of a retrospective study. Namely, there was no standardization of subjective and objective clinical measurements to measure patient progress during follow-up appointments following IA corticosteroid injection. Further, the patient population were cared by multiple providers who varied in documentation style, preference for ultrasound versus landmark guidance of IA injection, and follow-up. Therefore, some patients lacked adequate post-injection measurements. This lack of information reduced the overall number of patients included for analysis. Prospective study design would allow for standardization of pre-injection and post-injection measurements. Prospective cohort studies are needed in the future to overcome these limitations and determine predictive factors for treatment failure of IA corticosteroids for adhesive capsulitis.

## 5. Conclusions

IA corticosteroid injections for the management of adhesive capsulitis are associated with increased rates of failure and progression to surgical management when patients present with increased pain levels as well as with less improvement in pain levels ROM (e.g., ER in abduction) following injection.

## Figures and Tables

**Table 1 jcm-11-06212-t001:** Demographic and clinical categorical variables describing the study participants.

Variables	Levels	Frequency	Percentage
BMI (categorical)	>35	98	24.75
	≤35	298	75.25
Sex	Female	274	66.34
	Male	139	33.66
Race	African American	24	5.83
	American Indian, Eskimo, or Aleut	1	0.24
	Asian	5	1.21
	Hispanic	7	1.7
	Other	7	1.7
	White	368	89.32
Race (combined)	Non-white	44	10.68
	White	368	89.32
Diabetes	No	283	68.69
	Yes	129	31.31
Thyroid Disease	None	460	86.3
	Hypothyroid	69	12.95
	Hyperthyroid	4	0.75
Thyroid Disease	No	460	86.3
	Yes	73	13.7
Repeat injection?	No	418	78.42
	Yes	115	21.58
Surgery?	No	485	90.99
	Yes	48	9.01
Failure?	No	381	71.48
	Yes	152	28.52

**Table 2 jcm-11-06212-t002:** Demographic and clinical continuous variables describing the study participants.

Independent Variable	Levels	No Failure (n = 381)		Failure (n = 152)		*p*-Value
		N	%	N	%	
Pre-Forward Elevation	Mild restriction	36	32.73	16	32.00	0.9634
	Moderate restriction	54	49.09	24	48.00	
	Severe restriction	20	18.18	10	20.00	
Post-Forward Elevation	Mild restriction	51	68.00	22	56.41	0.0879
	Moderate restriction	22	29.33	12	30.77	
	Severe restriction	2	2.67	5	12.82	
Pre-Ext. Rot. (at the side)	Mild restriction	17	23.94	10	27.03	0.9269
	Moderate restriction	37	52.11	19	51.35	
	Severe restriction	17	23.94	8	21.62	
Post-Ext. Rot. (at the side)	Mild restriction	18	52.94	9	42.86	0.6997
	Moderate restriction	14	41.18	10	47.62	
	Severe restriction	2	5.88	2	9.52	
Pre-Ext. Rot. (in abduction)	Mild restriction	22	37.93	11	37.93	0.942
	Moderate restriction	32	55.17	17	58.62	
	Severe restriction	4	6.90	1	3.45	
Post-Ext. Rot. (in abduction)	Mild restriction	39	84.78	14	58.33	0.0311
	Moderate restriction	7	15.22	10	41.67	
	Severe restriction	---	---	---	---	

**Table 3 jcm-11-06212-t003:** Pre-injection and post-injection range of motion measurements in forward elevation, external rotation in abduction, and external rotation at the side.

ROM Measurement	Mild	Moderate	Severe
Pre-Forward Elevation	52 *(32.50%)*	78 *(48.75%)*	30 *(18.75%)*
Post-Forward Elevation	73 *(64.04%)*	34 *(29.82%)*	7 *(6.14%)*
Pre-External Rotation in Abduction	33 *(33.73%)*	49 *(56.32%)*	5 *(5.75%)*
Post-External Rotation in Abduction	53 *(75.71%)*	17 *(24.29%)*	0 *(0.00%)*
Pre-External Rotation at the Side	27 *(25.00%)*	56 *(51.85%)*	25 *(23.15%)*
Post-External Rotation at the Side	27 *(49.09%)*	24 *(43.64%)*	4 *(7.27%)*

Italics were used to indicate that these values were percentages.

**Table 4 jcm-11-06212-t004:** Treatment failure of IA injection based on severity of ROM restriction in forward elevation, external rotation in abduction, and external rotation at the side.

Independent Variable	Levels	No Failure (n = 381)		Failure (n = 152)		*p*-Value
		N	%	N	%	
Pre-Forward Elevation	Mild restriction	36	32.73	16	32.00	0.9634
	Moderate restriction	54	49.09	24	48.00	
	Severe restriction	20	18.18	10	20.00	
Post-Forward Elevation	Mild restriction	51	68.00	22	56.41	0.0879
	Moderate restriction	22	29.33	12	30.77	
	Severe restriction	2	2.67	5	12.82	
Pre-Ext. Rot. (at the side)	Mild restriction	17	23.94	10	27.03	0.9269
	Moderate restriction	37	52.11	19	51.35	
	Severe restriction	17	23.94	8	21.62	
Post-Ext. Rot. (at the side)	Mild restriction	18	52.94	9	42.86	0.6997
	Moderate restriction	14	41.18	10	47.62	
	Severe restriction	2	5.88	2	9.52	
Pre-Ext. Rot. (in abduction)	Mild restriction	22	37.93	11	37.93	0.942
	Moderate restriction	32	55.17	17	58.62	
	Severe restriction	4	6.90	1	3.45	
Post-Ext. Rot. (in abduction)	Mild restriction	39	84.78	14	58.33	0.0311
	Moderate restriction	7	15.22	10	41.67	
	Severe restriction	---	---	---	---	

## Data Availability

Not applicable.

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
