# Peer review of "Predictive Factors for Failure of Intraarticular Injection in Management of Adhesive Capsulitis of the Shoulder"

_jcm, 2022, doi:10.3390/jcm11206212_

Round 1

Reviewer 1 Report

Thank you for inviting me to review this interesting article. I have only a few queries.

In method part

Line 57, did the authors exclude other comorbid diseases such as subacromial bursitis, tendonitis, history of shoulder surgery, or SLAP in this study?

Line 60, it is kind to demonstrate the flowchart of the study about how many cases were screened, and how many were excluded (and based on which exclusion criteria).

Line 63, it is important for readers to understand the detail of ultrasound-guided injection method in the method part. Dosage? Which kind of corticosteroid? NSAIDs? With hydrodilation method?

Line 69, did the authors record the duration of frozen shoulder symptoms? In my opinion effects of IA injection could be different in different stages of frozen shoulder.

Result

In line 99 you mentioned the odds ratio but I cannot find the odds ratio by multivariate logistic regression results and table.

The tables were wrong, tables 3 and 4. Should be moved to 1 and 2?

Please check!

The discussion part is well written. I have no questions.

Reviewer 2 Report

Dear authors, thank you for the opportunity to review your manuscript regarding predictive factors for failure of intra-articular injection for adhesive capsulitis. It’s a worthy topic to investigate which has the potential to influence clinical practice.

A strength of the study is the relatively large sample size. Limitations undermine the usefulness of the study. The main finding that I found useful was that higher pre-intervention pain scores predict higher post-intervention pain scores.

I have made a few comments below that I hope you will find fair and useful.

Abstract:

Please update with my comments regarding the main manuscript in mind, excepting that you’ll be constrained by the word limit. For example, be explicit regarding what was compared and how (ie the stats).

Intro

Overall, this was concise and well written.

“current literature demonstrates that some patients do not experience the functional and pain-reducing benefits of IA corticosteroid injections” please state actual numbers, rather than “some”. This is more informative.

Hypotheses: I think that you should at least present information within the introduction to justify these hypotheses. For example, why do you think a history of diabetes would predict treatment failure? Obviously there’s on literature on this. Also, I think you need to be more specific with your hypothesis because stating that increased pain and poorer range of movement will result in increased rates of treatment failure seems like a circular argument to me ie treatment failure is ongoing pain and poor movement, so using these outcomes to predict poor outcomes, doesn’t seem helpful. Defining time points for your predictors in the hypothesis would be helpful, for example pre-procedure pain and poor movement.

Methods.

How does REDCap request information?

An accurate diagnosis is critical. Who diagnosed adhesive capsulitis? Inclusion criteria of pain and loss of range of motion seems a bit too vague for me, as lots of shoulder conditions can have these characteristics. How much pain and what was its location, and how much range of movement were considered necessary for the diagnosis? How were conditions such as symptomatic rotator cuff disease, bursitis and GHJ OA assessed for and excluded?

Including the NSAID group is a confounder. If they can’t be included in a subgroup analysis, my feeling is that they should be excluded from the study. Given the small number of NSAID and large number of corticosteroid injections, I think excluding the NSAID pts would simplify and strengthen the study.

Who gave the injection? Physician experience?

Who reviewed the charts?

On what type of pain scale was pain assessed? E.g VAS, 0-10 etc. Later, in the results, I see that you mention VAS. Include this information in the methods.

How was ROM assessed? E.g goniometer?

How long had pts had symptoms before the injection? Ie what stage of the disease were they likely to be in?

“initial 3 months after the injection” This a wide timeframe for assessing post-injection pain and ROM. Could it include day 1 versus day 90?

“Indications for subsequent injection and progression to surgical management were subjective in nature after incomplete resolution of symptoms following injection.” Obviously this is problematic and could result in wide variation in determining clinical failure and subsequent IA injection or surgery. It would be useful to know these pts post-procedure pain scores, ROM, satisfaction etc to see if they are any different from the group that didn’t proceed to AI injection or surgery.

“independent two-tailed t-test or Mann-Whitney” what was being compared?

Please indicate all the variables included in the analysis, how the variables were inputted and how it was determined whether to keep or exclude variables.

Results

Diabetes – type I or type II?

Table 2. p values – what stats are these based on? A footnote in the table might help in addition to explicit descriptions in the methods.

Also (and at the risk of being repetitive), using post injection ROM as a predictor of treatment failure seems confounded. Wouldn’t it have been used as a defining feature of treatment failure ie a predictor is also being used as a criteria for the outcome of interest. Using pre-op ROM seems useful as a predictor, not post-op ROM. I have the same sentiments for use of post IA injection pain scores.

“The ROM of each study participant in external rotation in abduction, external rotation at the side, and forward elevation was categorized by severity as mild, moderate, or 121 severe (Table 3).” Should this be table 2?

“For pre-injection external rotation in abduction, 33 (33.73%) had mild 122 restriction, 49 (56.32%) had moderate restriction, and 5 (5.75%) had severe restriction. For 123 post-injection external rotation in abduction, 53 (75.71%) had mild restriction, 17 (24.29%) 124 had moderate restriction, and 0 (0.00%) had severe restriction. For pre-injection external 125 rotation at the side, 27 (25.00%) had mild restriction, 56 (51.85%) had moderate restriction, 126 and 25 (23.15%) had severe restriction. For post-injection external rotation at the side, 27 127 (49.09%) had mild restriction, 24 (43.64%) had moderate restriction, and 4 (7.27%) had 128 severe restriction. For pre-injection forward elevation, 52 (32.50%) had mild restriction, 78 129 (48.75%) had moderate restriction, and 30 (18.75%) had severe restriction. For post-injec- 130 tion forward elevation, 73 (64.04%) had mild restriction, 34 (29.82%) had moderate re- 131 striction, and 7 (6.14 %) had severe restriction.”

There’s no point having a table if you present the data in the text. Use one or the other, not both.

Where are the results of the multivariate analysis?

Discussion

“This is a clinically impactful finding as it may allow physicians to provide improved patient-specific counseling as to likelihood of treatment success. It may improve patient expectations and decisions making when the injections do not hasten resolution.” I don’t find this statement particularly useful or believable when post intervention outcomes were included as predictors of post intervention outcomes. Using preop information to counsel pts of the likelihood of treatment success seems useful, and the results indicate that those with higher pain pre intervention tended to report higher pain post intervention. But counselling patients who have higher pain post intervention that they are more likely to have another injection or an operation seems to be intuitive and the results of this study probably don’t help with that conversation.

Para 2 is an interesting summary of the literature; however, the discussion should discuss the results of the current study in light of the existing literature. Please try and integrate your study findings within the literature within this para. Similar thoughts for para 3. For example, why do you think Zhang et al found an association between female and diabetes and IA injection but you didn’t?

“This current study found no significant differences in patient demographic features or medical comorbidities for treatment failure of IA corticosteroid in adhesive capsulitis.” This sentence simply restates the results. Discuss this result. What does it mean in light of your hypotheses and the literature etc? Diabetes should get a mention. What you did not find as significant is just as important as what you did find.

“some patients lacked adequate post-injection measurements.” Please bring this to the reader’s attention more explicitly in the results.

Conclusion

“IA corticosteroid injections for the management of adhesive capsulitis produce effective pain relief and functional improvements in the short- and mid-term.” This is not a conclusion from your study – focus on your conclusions.

Round 2

Reviewer 1 Report

The authors respond appropriately and no more comments

Reviewer 2 Report

Dear authors, thank you for considering my suggestions and updating the manuscript accordingly.